# DiCoDe: Diffusion-Compressed Deep Tokens for Autoregressive Video Generation with Language Models

## Abstract

Videos are inherently temporal sequences by their very nature. In this work, we explore the potential of modeling videos in a chronological and scalable manner with autoregressive (AR) language models, inspired by their success in natural language processing. We introduce **DiCoDe**, a novel approach that leverages **Di**ffusion-**Co**mpressed **De**ep Tokens to generate videos with a language model in an autoregressive manner. Unlike existing methods that employ low-level representations with limited compression rates, DiCoDe utilizes deep tokens with a considerable compression rate, approaching a 1000× reduction in token count when amortized over long chronological videos. This significant compression is made possible by a tokenizer trained through leveraging the prior knowledge of video diffusion models. Deep tokens enable DiCoDe to employ vanilla AR language models for video generation, akin to translating one visual "language" into another. By treating videos as temporal sequences, DiCoDe fully harnesses the capabilities of language models for autoregressive generation. DiCoDe is scalable using readily available AR architectures, and is capable of generating videos ranging from a few seconds to one minute using only 4 A100 GPUs for training. We evaluate DiCoDe both quantitatively and qualitatively, using generation quality mainly as evidence that such an aggressive amortized token compression remains usable for video synthesis under limited training resources. To showcase its scalability, we release a series of DiCoDe configurations with varying parameter sizes and observe a consistent improvement in performance as the model size increases from 100M to 3B. We believe that DiCoDe's exploration in academia represents a promising initial step toward scalable video modeling with AR language models, paving the way for the development of larger and more powerful video generation models.

## 1 Introduction

Autoregressive (AR) language models based on transformer architectures (Brown et al., 2020; Touvron et al., 2023; Le Scao et al., 2023; Chowdhery et al., 2023) such as GPT, with predicting the next token as the objective, have dominated generation tasks in natural language processing (NLP) while showcasing remarkable scalability (Kaplan et al., 2020). The unidirectional design of AR models aligns naturally with the sequential nature of language, where each token depends solely on its predecessors. Unlike other unidirectional models such as RNNs (Grossberg, 2013; Hochreiter & Schmidhuber, 1997), transformers exhibit greater scalability due to their parallel trainability and capability to handle longer contexts without strict Markovian constraints. This combination of unidirectional design and exceptional scalability makes AR models the preferred choice for generative tasks in NLP, where text is structured as a sequence of interconnected tokens.

*Similarly, videos can be viewed as sequential processes, akin to language.* However, prevailing methods (Blattmann et al., 2023; Chen et al., 2023; 2024b; Guo et al., 2023; Xing et al., 2025; Yang et al., 2024) for video generation often do not exploit this temporal characteristic. Instead, they tend to treat

---

This paper makes use of large language models (LLMs) exclusively for non-substantive assistance, including grammar correction, text formatting, LaTeX syntax support, and proofreading. All research ideas, mathematical formulations, experimental design, analysis, and conclusions are the original work of the authors.

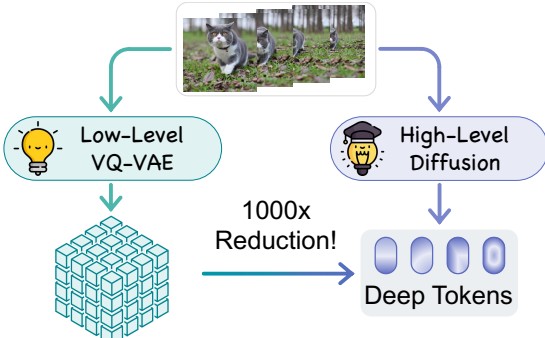

Figure 1: A VQ-VAE-based encoder (Yu et al., 2021) represents a 2-second $256 \times 256$ video segment with 16,384 low-level tokens. In DiCoDe, each sampled endpoint frame is represented by 16 high-level deep tokens; a standalone 2-second segment is decoded from the deep tokens of its first and last endpoint frames, while adjacent segments share endpoints in chronological long-video generation. Therefore, each additional segment introduces only 16 new deep tokens in the long-video amortized view, approaching a $1000\times$ reduction in token count.

videos as fixed-length clips generated simultaneously within a diffusion model (Rombach et al., 2022). While these approaches achieve satisfactory results for generating short clips, they do not accurately reflect the true nature of videos, leading to limitations in scalability, particularly when extending the time dimension. In scenarios involving abrupt scene transitions, these existing methods are likely to falter, underscoring the need for a more sophisticated approach that embraces the inherent temporal structure of videos for generation.

Consequently, the following question naturally arises: *Can videos be modeled in a chronological and scalable manner with autoregressive language models, replicating their success in NLP?* While the analogy between videos and language may seem straightforward at first glance, the redundancy inherent in video data poses a significant challenge. Recent work (Wang et al., 2024a) has explored the use of language models for autoregressive video generation by tokenizing video clips into low-level discrete tokens produced by a VQ-VAE (Yu et al., 2021)-based architecture, where a four-frame clip is represented by 4096 discrete codes. With this design, generating a minute-long video may necessitate a context window of up to a million tokens, which is obviously both infeasible and unaffordable. Therefore, there is a pressing need for a tokenizer that can condense video data into high-level tokens with a substantial compression rate.

In this work, we introduce **DiCoDe**, a novel approach that leverages **Di**ffusion-**Co**mpressed **De**ep Tokens to generate videos autoregressively with a language model in a chronological and scalable manner. By leveraging the prior knowledge of the video diffusion model (Xing et al., 2025), DiCoDe learns a frame-level tokenizer to encode temporally sampled endpoint frames into high-level tokens, trained through segment-level denoising. In DiCoDe, each sampled endpoint frame is represented by 16 continuous deep tokens. A standalone 2-second segment is decoded from the deep tokens of its first and last endpoint frames, whereas in a long chronological sequence adjacent segments share endpoints and each additional segment introduces only 16 new deep tokens. This gives an amortized compression rate approaching $1,000\times$ relative to low-level discrete tokens, making it feasible to model videos temporally with AR language models. These learned deep tokens essentially serve as a "language" for videos and are designed to satisfy the following properties: 1) **Temporally causal**: By encoding video clips in a way that preserves temporal order, DiCoDe aligns with the sequential nature of AR models and video data; 2) **Highly compressed**: By leveraging the prior knowledge of video diffusion model, videos can be represented with a manageable number of tokens for efficient AR modeling; 3) **Compatible with image data**: Our frame-level tokenizer allows images to be effectively represented, alleviating the scarcity of high-quality video-text data.

With this chronological and compact representation, DiCoDe employs vanilla AR language models for video generation. To fully unleash the scalability of AR models and utilize well-established architectures, DiCoDe does not rely on specialized designs like masking strategies with bidirectional attention in previous visual autoregression work (Li et al., 2024a; Xie et al., 2024). Instead, DiCoDe treats video generation as a

straightforward translation task, sequentially generating video tokens based on a text prompt given as the prefix. However, the original cross entropy loss in language models for training discrete tokens does not directly apply to continuous token modeling. Recent work (Li et al., 2024a) highlights that AR models are required to model the probability distribution for effective autoregression in a continuous-valued space. This necessity arises from the need to capture the variance of the data rather than relying solely on deterministic modeling. Inspired by this insight, we propose using a Gaussian Mixture Model (GMM) to model the uncertainty of the deep tokens, incorporating variance learning during the autoregressive process. The GMM modeling can be seamlessly integrated as a loss function with minimal modification to existing AR language models, providing a scalable and efficient solution for video generation.

We evaluate DiCoDe both quantitatively and qualitatively. First, we validate the video tokenization process, demonstrating that DiCoDe effectively compresses videos into high-level tokens with a significant compression rate and minimal quality degradation. We compare DiCoDe with established methods on zero-shot video generation tasks in terms of FVD (Unterthiner et al., 2018) and CLIPSIM (Wu et al., 2021; Radford et al., 2021), using these numbers to contextualize the cost-quality trade-off rather than to claim state-of-the-art visual quality. The experiment results show that DiCoDe remains functional as a video generator while using a compact AR sequence and limited computational resources (*i.e.*, 4 A100 GPUs). To further demonstrate its scalability, we train a series of DiCoDe configurations, ranging from 100M to 3B parameters, and observe a consistent performance improvement as the model size increases. The effectiveness of DiCoDe in video generation highlights the vast potential of AR models for temporally sequential video modeling. We hope that our promising initial step will draw increased attention to scalable video modeling using autoregressive language models.

## 2 Related Work

**Autoregressive Language Models in Visual Generation**   Autoregressive language models are currently emerging in both image and video generation. To match with language models, many methods (Wang et al., 2024a; Kondratyuk et al., 2023; Villegas et al., 2022; Yu et al., 2023; Esser et al., 2021; Ramesh et al., 2021; Ding et al., 2021; Yan et al., 2021; Ge et al., 2022; Li et al., 2024b; Wang et al., 2024b) tokenize visual data into discrete tokens using VQ-style tokenizers (Yu et al., 2021). Autoregressive video generation models such as MAGVIT-v2 (Yu et al., 2023) improve the discrete-token tokenizer itself, while VideoPoet (Kondratyuk et al., 2023), Emu3 (Wang et al., 2024a), Loong (Wang et al., 2024b), and Cosmos (NVIDIA et al., 2025) scale transformer-based sequence modeling to multimodal or world-modeling settings with compact video token representations. However, discrete raster-scan modeling still exposes video generation to long token sequences and step-by-step error accumulation. Recent work therefore explores alternative autoregressive formulations. NBP (Ren et al., 2025) predicts video blocks instead of individual tokens for faster semi-autoregressive decoding. NOVA (Deng et al., 2025) removes vector quantization and combines temporal frame-by-frame prediction with spatial set-by-set prediction. FAR (Gu et al., 2025) studies long-context frame-level autoregressive video modeling with next-frame prediction. VideoMAR (Yu et al., 2025) further studies continuous-token video AR with temporal causality and spatial masked generation, while MAGI-1 (Sand.ai et al., 2025) scales causal chunk-wise video generation with a large autoregressive system. These developments indicate that autoregressive video generation is an active direction, but most of them either retain dense visual token maps, introduce specialized block/mask/chunk objectives, or rely on substantially larger training setups. DiCoDe takes a complementary route: it uses a video diffusion model as a high-ratio compressor, producing a very short sequence of deep continuous tokens so that a largely vanilla AR language model can be used for temporal modeling.

**Continuous Tokens for Visual Autoregression**   Another line of work studies whether visual autoregression must rely on discrete tokens. GIVT (Tschannen et al., 2024) and MAR (Li et al., 2024a) show that continuous visual tokens can be modeled autoregressively when the model predicts a distribution rather than a deterministic point. Recent extensions and variants further bridge continuous and discrete representations or adapt continuous-token objectives to video (Wang et al., 2025; Yu et al., 2025; Deng et al., 2025; Feng et al., 2025). Different from these methods, which generally operate on image or video VAE latents, DiCoDe learns deep tokens as conditions for a video diffusion model. The resulting representation is not meant to

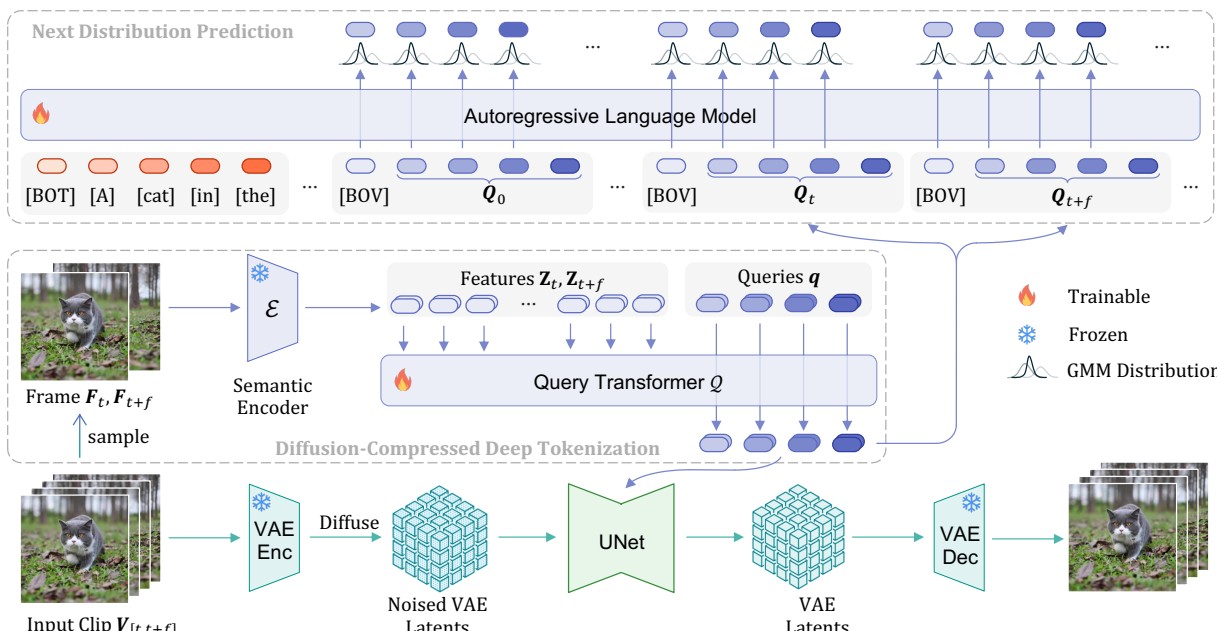

Figure 2: The overall framework of DiCoDe, which consists of a video diffusion model as the tokenizer to extract highly-compressed deep tokens and an autoregressive language model to predict the sequence of deep tokens through modeling distributions.

preserve all low-level visual details in the token sequence; instead, it delegates local video synthesis to the diffusion tokenizer and reserves the AR model for compact temporal planning.

**Video Diffusion Models**  Diffusion models are the prevalent method for video generation. Most diffusion models utilize 3D U-Net or extend the UNet in T2I models with temporal layers. (Ho et al., 2022b;a; Singer et al., 2022; Chen et al., 2023; Zhou et al., 2022; Wang et al., 2023b; Blattmann et al., 2023; Guo et al., 2023; Zeng et al., 2024) Recently, DiT (Peebles & Xie, 2023)-based diffusion models show promising results in video generation (Ma et al., 2024; Yang et al., 2024; Kong et al., 2024; Lin et al., 2024; Team Wan et al., 2025). Diffusion models are trained on fixed-length video clips like 16 frames. Despite diffusion models can be used in autoregressive or sliding-window manners to generate long videos (Henschel et al., 2024; Qiu et al., 2023; Lu et al., 2024; Kim et al., 2024; Chen et al., 2024a; Yin et al., 2025), they are limited by the receptive field and encounter consistency issues in long video generation. Recent large-scale systems also incorporate causal or chunk-wise denoising to improve temporal extension (Sand.ai et al., 2025). In our work, we use a fixed-length diffusion model differently: rather than relying on it as the main long-horizon generator, we use it as a high-ratio compressor with prior world knowledge, and leave global dynamics and consistency to the autoregressive language model.

## 3  Method

As illustrated in Fig. 2, DiCoDe is composed of a video diffusion model that serves as the tokenizer for extracting deep tokens and an autoregressive language model that predicts sequences of deep tokens. We first explain the rationale behind the design of deep tokens in Sec. 3.1. In Sec. 3.2, we describe how video diffusion models are utilized to learn the deep tokens. In Sec. 3.3, we describe how vanilla autoregressive language models are used to model sequences of deep tokens. Finally, in Sec. 3.4, probabilistic modeling with GMMs is introduced to facilitate the learning of AR models over continuous tokens.

### 3.1 Designing a Language for Videos

This "language", i.e. deep tokens, serves as a proxy between the video diffusion model and the AR model. We justify the design of deep tokens from both cognitive and practical perspectives. The following principles must be adhered to: **i)** Sequential in time. To match the sequential nature of AR models and video data, deep tokens must be chronological, which is crucial for naturally extending to longer videos and enhancing scalability. **ii)** Highly compressed. While an adult reads 200-300 words per minute in English (Brysbaert, 2019), a one-minute $256 \times 256$ video of 8FPS typically requires 8FPS $\times$ 256Tokens / Frame $\times 60s \approx 120$k tokens to model, which is obviously unmanageable for AR models. From a cognitive point of view, modeling videos analogy to language requires an extremely compressed tokenization method. **iii)** Compatible with image data. High-quality video-text data is scarce compared to image-text data, with commonly used datasets like WebVid (Bain et al., 2021) are significantly smaller in magnitude than image-text datasets such as LAION (Schuhmann et al., 2022). However, autoregressive models are data-hungry. To mitigate the scarcity of video-text data, deep tokens should integrate seamlessly with image data, allowing the model to leverage the abundant image-text data for much richer semantic information.

Given a video of varying length $\mathbf{V} \in \mathbb{R}^{T \times H \times W \times 3}$, DiCoDe first samples endpoint frames with a fixed temporal interval $f$, achieving an $f\times$ reduction along the temporal axis for the AR sequence. The sampled endpoint frames are denoted as $\mathbf{F}_0, \mathbf{F}_f, \ldots, \mathbf{F}_{\lfloor T/f \rfloor f}$. The endpoint frames are then encoded into high-level semantic-rich tokens individually using an image encoder $\mathcal{E}$ as features $\mathbf{Z}_t \in \mathbb{R}^{N \times C}$, where $t$ is the frame index, $N$ is the number of encoder tokens, and $C$ is the channel dimension. Although $N$ is usually already smaller than the number of typical low-level tokens, it is still too large for long-context AR modeling when $T$ is large. The high-level tokens are further compressed into deep tokens using a query transformer $\mathcal{Q}$. Specifically, $\mathcal{Q}$ learns a fixed set of $N_q$ queries $\mathbf{q} \in \mathbb{R}^{N_q \times C}$ by concatenating $[\mathbf{q}, \mathbf{Z}_t]$ and passing them through a multi-layer self-attention transformer. The final output of $\mathcal{Q}$ is a set of $N_q$ high-level continuous tokens $\mathbf{Q}_t \in \mathbb{R}^{N_q \times C}$, which are used as the deep tokens for endpoint frame $t$.

This design encodes videos in a frame-wise manner and is therefore chronological and compatible with image data. Temporal endpoint sampling and query-based token compression together achieve an extremely high amortized compression rate for long videos. Encoder $\mathcal{E}$ and query transformer $\mathcal{Q}$ construct a mapping $p(q|v)$ from data space $\mathbf{V} \sim p_{\text{data}}(v)$ to a high-level continuous space $\mathbf{Q} \sim p_{\text{high}}(q)$. DiCoDe learns this mapping via video diffusion models.

### 3.2 Video Diffusion Models as the Tokenizer

Conventional tokenziers like VAE, VQ-VAE, or VQ-GAN cannot achieve a such high compression rate. DiCoDe turns to video diffusion models as the tokenizer with rich prior knowledge for an extremely high compression ratio. The frame-level encoding and clip-level decoding of deep tokens can meet the design principles and achieve satisfying reconstruction quality at the same time.

Given two consecutive sets of endpoint deep tokens $\mathbf{Q}_s$ and $\mathbf{Q}_e$, where $e = s + f$, we train a video diffusion model to reconstruct the ground-truth segment $\mathbf{V}_{[s,e]} \in \mathbb{R}^{(f+1) \times H \times W \times 3}$. Here $f$ denotes the number of frame intervals, so the segment includes both endpoints $\mathbf{F}_s$ and $\mathbf{F}_e$; in our default setting, $f = 16$ and the model reconstructs a 17-frame segment spanning 2 seconds at 8 fps. Let $\epsilon \sim \mathcal{N}(\mathbf{0}, \mathbf{I})$ be Gaussian noise and $\mathbf{V}_{[s,e],t}$ be the noised segment at denoising timestep $t$. Following the v-prediction parameterization used in training, we define the velocity target as $\mathbf{v}_t = \alpha_t \epsilon - \sigma_t \mathbf{V}_{[s,e]}$, where $\alpha_t$ and $\sigma_t$ are the scheduler coefficients. The diffusion loss is defined as

$$\mathcal{L}_{\text{diff}}(\mathbf{V}_{[s,e]}, \mathbf{Q}_s, \mathbf{Q}_e) = \mathbb{E}_{\epsilon,t}\left[\left\|\mathbf{v}_t - \mathbf{v}_\theta(\mathbf{V}_{[s,e],t}, t, \mathbf{Q}_s, \mathbf{Q}_e)\right\|^2\right], \tag{1}$$

where $\mathbf{v}_\theta$ is the prediction of the diffusion model with learnable parameters $\theta$ conditioned on the noised input $\mathbf{V}_{[s,e],t}$, denoising timestep $t$, and endpoint deep tokens $\mathbf{Q}_s$ and $\mathbf{Q}_e$.

The mapping $p(v|q)$ essentially reconstructs a short video segment from the deep tokens of its first and last endpoint frames. We make an assumption that video is sufficiently redundant for a short segment (e.g., 2 seconds) to be reconstructed from compact endpoint representations. This assumption is required by the temporal downsampling design in Sec. 3.1 and can be satisfied with a powerful video diffusion model

leveraging its world knowledge from massive pre-training data. We validate this assumption in Sec. 4.3. Except for the endpoint deep tokens used by the tokenizer, DiCoDe does not apply text prompts or preceding-frame conditions to the video diffusion model, even though such conditions may improve performance. We employ the video diffusion model as a high-level tokenizer and leave chronological generation to the AR model.

### 3.3 Modeling Videos with Autoregressive Models

To make the most of the scalability and mature techniques of AR language models, DiCoDe employs a vanilla autoregressive transformer for video generation. Let $\mathbf{T}$ denote the conditional text tokens and let $\mathbf{Q}_n = \{Q_{(n,0)}, \dots, Q_{(n,N_q-1)}\}$ denote the $N_q$ deep tokens of the $n$-th sampled endpoint frame. We add a [BOV] delimiter token at the beginning of each sampled endpoint frame. The AR model follows the temporal order of endpoint frames and the token order within each frame, and factorizes the conditional likelihood as

$$p(\mathbf{Q}_{0:N_f-1} \mid \mathbf{T}) = \prod_{n=0}^{N_f-1} \prod_{m=0}^{N_q-1} p\left(Q_{(n,m)} \mid \mathbf{T}, \mathbf{Q}_{<n,m}\right), \tag{2}$$

where $N_f$ is the number of sampled endpoint frames and $\mathbf{Q}_{<n,m}$ contains all previously generated visual tokens and the corresponding [BOV] delimiters before $Q_{(n,m)}$.

DiCoDe does not apply any special design for deep tokens such as masking or bidirectional modeling, which are found not helpful in our settings. We attribute this to the compact nature of deep tokens. Unlike VAE latents or discrete tokens, deep tokens are already highly compressed and semantically rich, thus do not rely heavily on each other for reconstruction. DiCoDe treats video generation as a translation task. This simple choice of AR models allows the utilization of readily available AR architectures and pre-trained models, even if they are designed for text data.

### 3.4 Explicitly Introducing Conditional Uncertainty

Due to the decoupled and offline nature of the diffusion tokenizer, the AR model receives continuous point targets extracted by the tokenizer. Unlike the discrete-token setting, where cross entropy naturally models a categorical distribution, a naive continuous-token objective such as $\mathcal{L}_2$ corresponds to a fixed-variance Gaussian likelihood centered at the predicted mean. Specifically, minimizing $\|q - \mu\|^2$ is equivalent, up to constants and a fixed scale, to minimizing the negative log-likelihood of $q$ under $\mathcal{N}(\mu, \sigma^2\mathbf{I})$ with fixed $\sigma$. This fixed-variance unimodal assumption is too restrictive for autoregressive modeling of high-level continuous deep tokens, where multiple future visual states may be plausible under the same prefix. As found in Li et al. (2024a), directly employing an $\mathcal{L}_2$ loss performs poorly for continuous visual autoregression.

Instead of directly predicting a single point estimate for $\mathbf{Q}$, DiCoDe predicts the parameters $\mathcal{P}$ of a conditional distribution $p_\mathcal{P}(q \mid \mathbf{T}, \mathbf{Q}_{<n,m})$. During training, the loss is the negative log-likelihood of the target token under the predicted distribution,

$$\mathcal{L}_{\text{NLL}} = -\log p_\mathcal{P}\left(Q_{(n,m)} \mid \mathbf{T}, \mathbf{Q}_{<n,m}\right). \tag{3}$$

At inference, DiCoDe samples from the predicted distribution to generate the next deep token.

The choice of the predefined conditional distribution is flexible. We investigate two types of distributions, including a diagonal Gaussian model and a Gaussian Mixture Model (GMM), referred to as Gaussian loss and GMM loss, respectively. For a simple Gaussian model, DiCoDe predicts the mean $\mu_{(n,m)}$ and standard deviation $\sigma_{(n,m)}$ for each token $Q_{(n,m)}$. For a $K$-component GMM, DiCoDe predicts component weights $\pi_{(n,m),k}$, means $\mu_{(n,m),k}$, and scale parameters for each component. For numerical stability, the AR head predicts unconstrained scale parameters, which are transformed into positive diagonal variances by a softplus function. We denote the resulting standard deviations by $\sigma_{(n,m),k}$ and define

$$p_\mathcal{P}\left(q \mid \mathbf{T}, \mathbf{Q}_{<n,m}\right) = \sum_{k=1}^{K} \pi_{(n,m),k}\, \mathcal{N}\left(q \mid \mu_{(n,m),k}, \text{diag}(\sigma^2_{(n,m),k})\right). \tag{4}$$

Table 1: Datasets used for training the AR language models. Only a small portion of image data (a total of 25M) is used.

| Name | Type | Num Pairs |
|---|---|---|
| JourneyDB (Sun et al., 2023) | Image-Text | 4M |
| Unsplash (Unsplash, 2024) | Image-Text | 2M |
| LAION-Aesthetics v2 (score $\geq$ 4.5) (Schuhmann et al., 2022) | Image-Text | 1.2B |
| LAION-COCO (Schuhmann et al., 2022) | Image-Text | 600M |
| WebVid-10M (Bain et al., 2021) | Video-Text | 10M |

The training loss itself is computed analytically by the GMM NLL with a log-sum-exp over mixture components. When sampled continuous tokens are used as model inputs during training, the continuous Gaussian noise within a selected component is sampled with the reparameterization trick. At inference, a component is sampled according to the predicted mixture weights, and the next token is sampled from that component. We provide an ablation study on the choice of the target distribution in Section 4.5, demonstrating the impact of different distributional choices on the performance of DiCoDe.

## 4 Experiments

### 4.1 Implementation Details

**Architecture.** The tokenizer can be implemented with any conditional video generation models. We use the off-the-shelf video diffusion model DynamiCrafter (Xing et al., 2025). DynamiCrafter is a triple-conditioned model conditioned with a text prompt, global visual condition, and full image condition. To align with the design of DiCoDe, we remove the text prompt and full image condition, and learn the deep tokens as the global visual condition. We start from the pre-trained $256 \times 256$ DynamiCrafter model, change the first convolution from 8 to 4 input channels, and reconstruct 17-frame video segments spanning 2 seconds at 8 fps. Except for the ablation study, each sampled endpoint frame is represented by 16 1024-dimensional deep tokens. Thus, a standalone 2-second diffusion segment is conditioned on 32 endpoint deep tokens in total, while chronological long-video generation introduces 16 new deep tokens for each additional segment after endpoint sharing. CLIP-ViT-H/14 trained with OpenCLIP on LAION-2B (Radford et al., 2021; Cherti et al., 2023) is used as the semantic encoder, followed by the 4-layer, 12-head query transformer inherited from DynamiCrafter; in the 16-token setting, we keep the first 16 learned queries. Benefiting from the design of deep tokens, we can leverage existing powerful language models as AR models. To demonstrate language knowledge transferability, we use two families of pre-trained AR models: GPT-2 (Radford et al., 2019) and Llama 3.2 (Meta, 2024; Grattafiori et al., 2024). Different sizes of models are used to demonstrate the scalability of DiCoDe. For GPT-2, we use GPT-2 (117M), GPT-2 Medium (345M), and GPT-2 Large (774M). For Llama 3.2, we use Llama 3.2-1B (1.23B) and Llama 3.2-3B (3.21B). We use the language models' original tokenizers and pad or truncate each text prompt to 80 tokens. Each visual frame is represented by a [BOV] token, 16 deep tokens, frame-level positional embeddings, and token-level positional embeddings; two 3-layer MLPs project between the visual-token dimension and the language-model hidden dimension. For GMM loss, we use 16 components, so the final projection layer of the AR model predicts $16 \times 1024 \times 2 + 16 = 32784$-dimensional features for the component means, variance parameters, and mixture logits of each deep token. We use DiCoDe-Llama3.2-1B as the default model unless otherwise specified.

**Datasets.** For the tokenizer, we use the WebVid-10M dataset with 10M video clips. For the AR models, we use a mix of image and video datasets for richer knowledge listed in Tab. 1. We use the mix of large-scale of datasets for diversity and only randomly sample *a small portion* of the samples (25M) without repetition since we are already using powerful pre-trained models and limited by the computational resources. For video data, clips are resized and center-cropped to $256 \times 256$, sampled as 17-frame segments with a random frame stride from 1 to 6 for the tokenizer, and temporally downsampled to 0.5 FPS for AR training. We also filter a subset of high-motion WebVid videos using optical flow: the average flow magnitude between

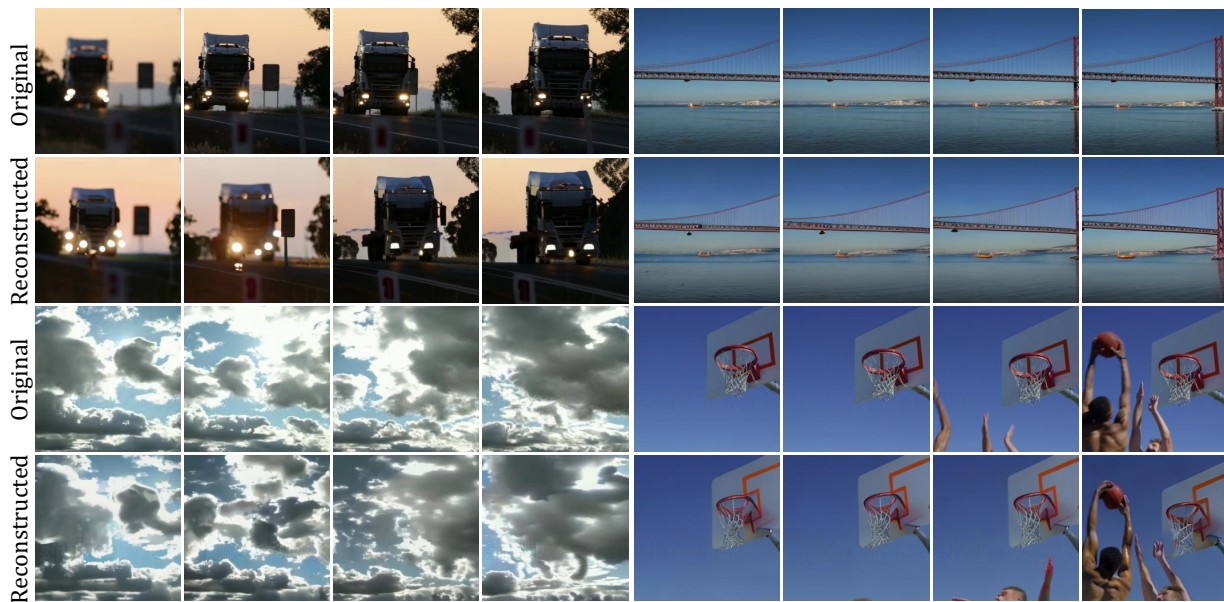

Figure 3: Results of tokenization. A 17-frame video segment spanning 2 seconds can be reconstructed effectively from the endpoint representations of its first and last frames, with 16 deep tokens per endpoint frame, even with object motion (left-top), camera motion (right-top), complex scenes (left-bottom), and emerging entities (right-bottom).

Table 2: Zero-shot video generation results on MSR-VTT dataset.

| Model | CLIPSIM ↑ | FVD ↓ |
|---|---|---|
| CogVideo (Hong et al., 2022) | 0.2631 | 1294 |
| ModelScopeT2V (Wang et al., 2023a) | 0.2930 | 550 |
| Show-1 (Zhang et al., 2024) | **0.3072** | 538 |
| VideoPoet (Kondratyuk et al., 2023) | 0.3049 | **213** |
| Loong (Wang et al., 2024b) | 0.2903 | 274 |
| CAT-LVDM (Maduabuchi et al., 2025) | – | 396.4 |
| DiCoDe-Llama3.2-1B | 0.2950 | 367 |

consecutive 1 FPS frames is used as a motion score, and videos with score in $[0.5, 1.38]$, length between 556 and 4000 frames, minimum aspect ratio 0.3333, and minimum resolution 200 are retained.

**Training.** For the training of video diffusion model, we train on WebVid-10M for 100k iterations with a batch size of 64, using a fixed learning rate of 1e-5, v-prediction and AdamW (Loshchilov & Hutter, 2019) optimizer, after removing WebVid watermarks using template matching. For the training of AR models, we train in a progressive fashion for faster convergence. The model is first trained on image-text datasets for 100k iterations with $\mathcal{L}_2$ loss and then for another 100k iterations with GMM loss. It is then trained on mixed image and video data for 100k iterations with an image:video ratio of 1:1, followed by 20k iterations on the filtered motion videos. For each AR training video, we sample a 32-second span and keep 16 sampled endpoint frames at 0.5 FPS, giving a maximum AR context length of $80 + (1 + 16) \times 16 = 352$ tokens including text tokens, [BOV] delimiters, and deep tokens. We use a global batch size of 256, cosine-scheduled learning rate starting at 1e-4 with warm-up for 1k iterations, AdamW optimizer with $\beta_1 = 0.9$, $\beta_2 = 0.98$, $\epsilon = 1e-6$, and weight decay 0.05, and 5% visual-token dropout for classifier-free guidance. After AR training, the video diffusion model is further fine-tuned for 20k iterations using predicted deep tokens to reduce the train-test mismatch introduced by probabilistic AR sampling. At inference, the video diffusion model uses 50 DDIM sampling steps, CFG (classifier-free guidance) scale of 7.5, and guidance rescale of 0.7.

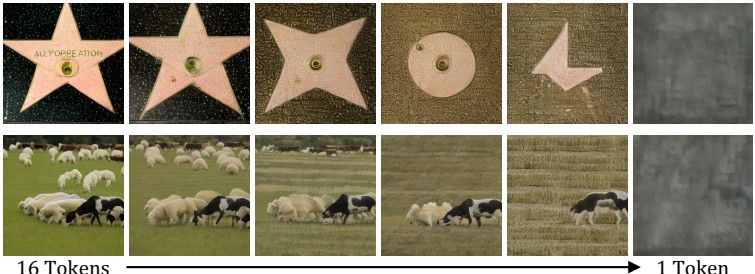

16 Tokens ⟶ 1 Token

Figure 4: Zeroing out deep tokens during tokenization reduces the number of entities in reconstructed frames.

## 4.2 Quantitative Results

To provide a quantitative evaluation of DiCoDe, we evaluate the zero-shot video generation task on the MSR-VTT (Xu et al., 2016) dataset and show the results in Tab. 2. We use the full set of 2990 test videos and randomly sample one caption for each video. We report CLIPSIM (Wu et al., 2021; Radford et al., 2021) and FVD (Unterthiner et al., 2018) of DiCoDe-Llama 3.2-1B as the metrics for comparison. These results should be read as a reference point for the compression-efficiency trade-off of DiCoDe, since the compared systems differ substantially in data scale, model scale, conditioning design, and evaluation protocols.

In this 17-frame, 2-second segment setting, a standalone diffusion segment uses 32 endpoint deep tokens for a $256 \times 256$ video segment. In chronological long-video generation, adjacent segments share endpoints, so each additional segment introduces only 16 new deep tokens. In contrast to our highly compressed design, other methods utilize orders of magnitude more visual tokens. For example, CogVideo (Hong et al., 2022) requires 6400 tokens at a spatial resolution of 160x160, Loong requires 1024 tokens at 128x128, and VideoPoet requires 1280 tokens at 128x128. Other methods also have been pre-trained on massive image-text datasets or use powerful pre-trained text-to-image models. For example, Show-1 (Zhang et al., 2024) utilizes DeepFloyd as initialization and VideoPoet (Kondratyuk et al., 2023) is trained on 1B image-text pairs. DiCoDe, on the other hand, is trained on a subset of 25M image-text pairs without repetition. And the diffusion tokenizer is only conditioned on deep tokens without text conditioning.

Despite the highly compressed design and limited training, DiCoDe produces reasonable quantitative results for a compact-token AR video generator. Unlike other methods that have sufficient image-text pre-training, DiCoDe is trained on image data at a relatively smaller scale. The table therefore mainly shows that the proposed representation remains usable after aggressive compression, while leaving large-scale visual-quality optimization as an orthogonal direction.

## 4.3 Qualitative Results

**Tokenization** The effectiveness of the diffusion-powered tokenization is shown in Fig. 3. The reconstructed 17-frame, 2-second videos are generated from compact endpoint representations, with 16 deep tokens for the first endpoint frame and 16 deep tokens for the last endpoint frame. Despite the extremely high amortized compression ratio, DiCoDe successfully reconstructs the video segments with minimal degradation. The results confirm our hypothesis that short video segments are sufficiently redundant to be represented with a few endpoint deep tokens, even with object motion, camera motion, complex scenes, and emerging entities. To investigate the essence of deep tokens, we gradually zero out deep tokens and show the results in Fig. 4. Removing deep tokens reduces the entities in the video one by one, indicating their rich and condensed semantics, akin to language. This behavior suggests that the learned tokens encode scene-level factors rather than simply serving as low-level reconstruction codes.

**Short Video Generation** We qualitatively compare the short video generation results of DiCoDe with other autoregressive methods with examples in Fig. 5.

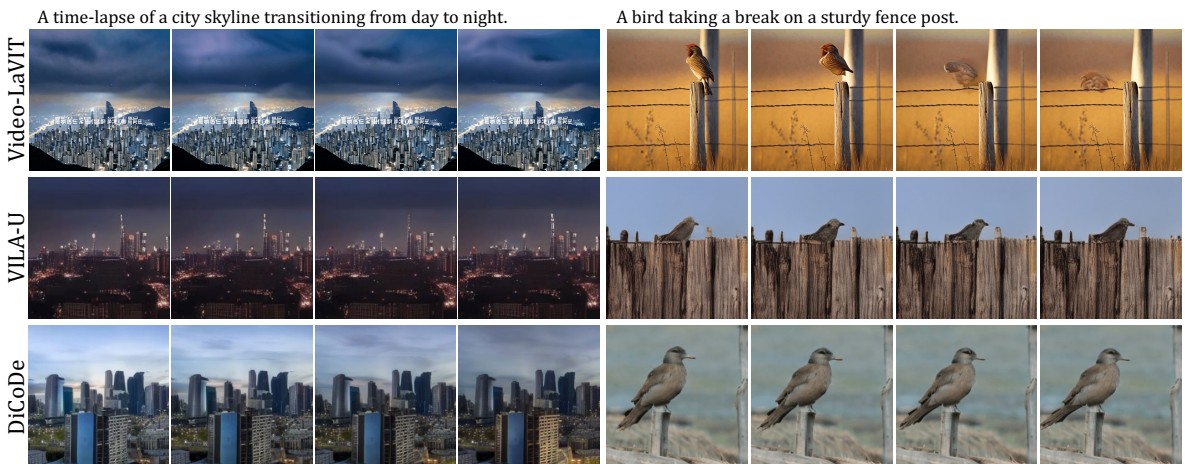

Figure 5: Comparison of short video generation results. DiCoDe generates more coherent, dynamic and prompt-following videos. On the left, it captures the transition from day to night. On the right, it maintains the appearance of the bird across frames with noticeable motion.

An elderly couple walking hand in hand, surrounded by a sunset's glow.

A black and white photograph of an old train traveling through the countryside.

An animation of a hot air balloon .

A dramatic sunset over a calm sea.

Figure 6: Additional results generated by DiCoDe.

A close-up of a butterfly landing on a flower.

A joyful girl dancing freely on the beach, moving in rhythm with the ocean's waves.

A digital animation of a robot exploring a futuristic city.

Figure 7: Long videos generated by DiCoDe. Frames are evenly sampled from 256-frame (64s) videos.

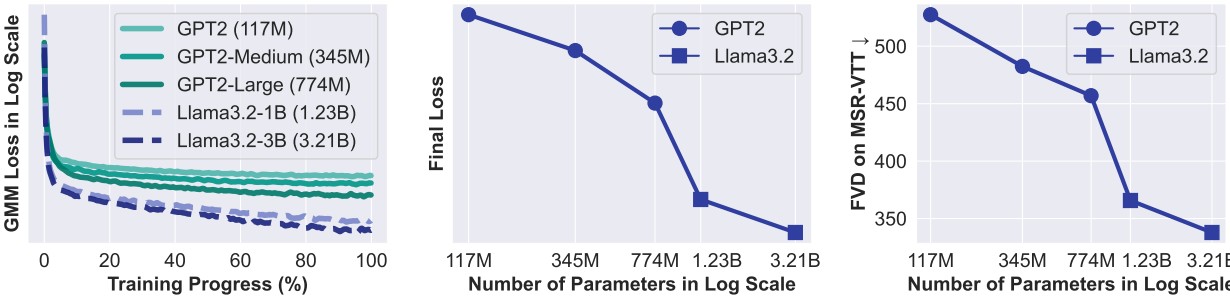

Figure 8: Ablation on AR model sizes. Larger models achieve lower loss and FVD, demonstrating clear scalability.

Video-LaVIT (Jin et al., 2024) heavily relies on the pre-trained text-to-image model for visual quality but finds it hard to fully capture motion dynamics. It generates almost static images for time-lapse transition or cannot maintain object appearance. VILA-U (Wu et al., 2024) also fails to capture the transition from day to night and generates deformed objects. In these examples, DiCoDe generates coherent, dynamic and prompt-following videos. It captures the transition from day to night, preserves the appearance of moving objects across frames, and avoids the nearly static outputs observed in several comparison examples. The additional comparisons in Fig. 9 show a similar pattern across hand motion, sunset transition, boat motion, and butterfly motion: the compared AR baselines often deform hands and objects or collapse into nearly static clips, while DiCoDe better preserves object consistency and visible temporal change. More results are illustrated in Fig. 6.

**Long Video Generation** In Fig. 7, we show the long video generation results of DiCoDe. DiCoDe can extend generation to longer videos with largely consistent appearance and motion dynamics. Compared to other autoregressive long video generation methods like Loong (Wang et al., 2024b), DiCoDe does not require additional techniques like truncating or prefixing for long video extension, since the deep-token sequence remains short. Constrained by data and computational resources, we sample 256 output frames at 4 fps, corresponding to 64 seconds for each video. At the output-video level, these samples contain 256 visual time steps; the AR model itself operates on the much shorter sequence of temporally sampled endpoint deep tokens, leaving substantial room within the context length of modern language models and demonstrating the potential for extending longer.

### 4.4 Scalability of Autoregressive Language Models

Scalability is the key advantage of employing vanilla AR language models. We verify the scalability of DiCoDe across different sizes of AR models in terms of training loss and reconstruction quality. All experiments use the same setting except for the size of the AR models. As listed in Fig. 8, given the same training budget, larger models achieve lower pre-training loss and lower FVD. There is a clear gap between GPT2 family and Llama3.2 family, which is attributed to the better pre-training of Llama3.2. This gap confirms that even across modality, the knowledge from text can be transferred to video generation to some extent, showing that video generation is indeed treated as a language generation with the deep tokens design. The Llama3.2 models are also far from convergence under our training budget, indicating potential for further improvement with more training.

### 4.5 Ablation Studies

**Number of Deep Tokens** We ablate the effect of the number of deep tokens per endpoint frame in tokenization, including 8, 16, and 32 tokens. Tokenizers with different numbers of endpoint-frame tokens are first trained with the same settings. Then, the tokenizers are used to generate videos with Llama 3.2-1B. Qualitative results show that more tokens lead to better reconstruction quality. The zero-shot generation results are shown in Tab. 3. The FVD is reported after image-video mixed training for ablation

Table 3: Ablation study on the number of deep tokens per endpoint frame, evaluated with zero-shot video generation.

| Num Deep Tokens per Endpoint Frame | 8 | 16 | 32 |
|---|---|---|---|
| FVD on MSR-VTT ↓ | 600 | **565** | 641 |

Table 4: Ablation study on the loss type for training the language model, evaluated with zero-shot video generation.

| Loss Type | $L_2$ | Gaussian | 16-GMM |
|---|---|---|---|
| FVD on MSR-VTT ↓ | 643 | 593 | **565** |

purposes. The 16-token tokenizer achieves the best FVD, while the 32-token setting performs worse than the 8-token setting. This aligns with the compact-token design: longer AR visual sequences are more prone to accumulated prediction error, and the 32-token setting increases the AR sequence length relative to the 16-token setting. Therefore, for best performance, one needs to strike a balance between reconstruction quality and compression ratio.

**Loss Type** Conditional distribution modeling is a key design of DiCoDe. We ablate different loss types for the AR models, including $\mathcal{L}_2$ loss, Gaussian loss, and GMM loss with 16 components. The results are shown in Tab. 4. The GMM loss performs the best while the $\mathcal{L}_2$ loss performs the worst. This is expected because the GMM loss can better capture conditional uncertainty and multi-modality in the distribution of deep tokens, which is important for diverse generation.

## 5 Conclusion

We propose DiCoDe, a video generation framework that models videos in a chronological and scalable manner with autoregressive language models. With the design of deep tokens, diffusion-powered compression, and probabilistic modeling of continuous targets, DiCoDe generates videos as temporal sequences, aligning with their sequential nature. The experiments demonstrate the effectiveness and scalability of DiCoDe. We hope DiCoDe reveals a promising paradigm for video generation and inspires the development of larger-scale long video generation models in the future.

**Statement of Broader Impact.** Since our framework advances the capabilities of generative video synthesis, it shares the broader societal implications associated with deep generative models, such as the potential for misuse in creating misleading content, deepfakes, or copyright infringement. Furthermore, as our method builds on large-scale video data and generative priors, there is a risk of propagating or amplifying biases inherent in the underlying datasets and foundation models.

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

---

**Algorithm 1** Gaussian Mixture Model Loss Computation

---

**Require:** Means $\boldsymbol{\mu}$ of size $B \times K \times d$, variances $\boldsymbol{\sigma}^2$ of size $B \times K \times d$, mixture logits $\mathbf{a}$ of size $B \times K$, target tokens $\mathbf{X}$ of size $B \times d$
**Ensure:** Negative log-likelihood loss nll
 1: **function** GMM Loss($\boldsymbol{\mu}$, $\boldsymbol{\sigma}^2$, $\mathbf{a}$, $\mathbf{X}$)
 2:     $\log \boldsymbol{\pi} \leftarrow \log\_\text{softmax}(\mathbf{a}, \dim = K)$
 3:     **for** $b = 1$ to $B$ **do**
 4:         **for** $k = 1$ to $K$ **do**
 5:             $\ell_{b,k} \leftarrow \log \pi_{b,k} - \frac{1}{2} \sum_{j=1}^{d} \left[ \log(2\pi\sigma_{b,k,j}^2) + \frac{(X_{b,j} - \mu_{b,k,j})^2}{\sigma_{b,k,j}^2} \right]$
 6:         **end for**
 7:         $L_b \leftarrow \text{logsumexp}_k(\ell_{b,k})$
 8:     **end for**
 9:     nll $\leftarrow -\frac{1}{B} \sum_{b=1}^{B} L_b$
10:     **return** nll
11: **end function**

---

# A  Implementation Details

**Gaussian Mixture Model Loss**  We use 16 components for the GMM loss by modifying the final projection layer in the autoregressive language models. For each deep token, the model predicts component means, variance parameters, and mixture logits. The algorithm for computing the GMM loss is shown in Alg. 1.

# B  Additional Qualitative Results

The videos showcasing qualitative results are available in the attached files. Additionally, Fig. 10 and Fig. 11 present further results on short and long video generation, respectively. The outputs from DiCoDe are dynamic, coherent, and faithfully adhere to the provided prompts.

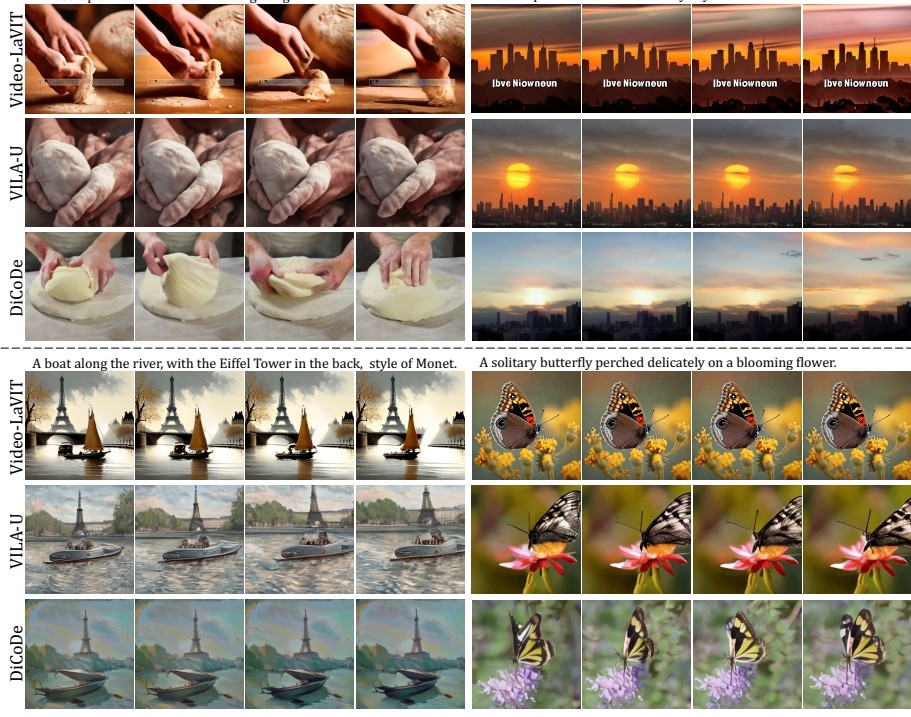

Figure 9: Additional comparison of short video generation results.

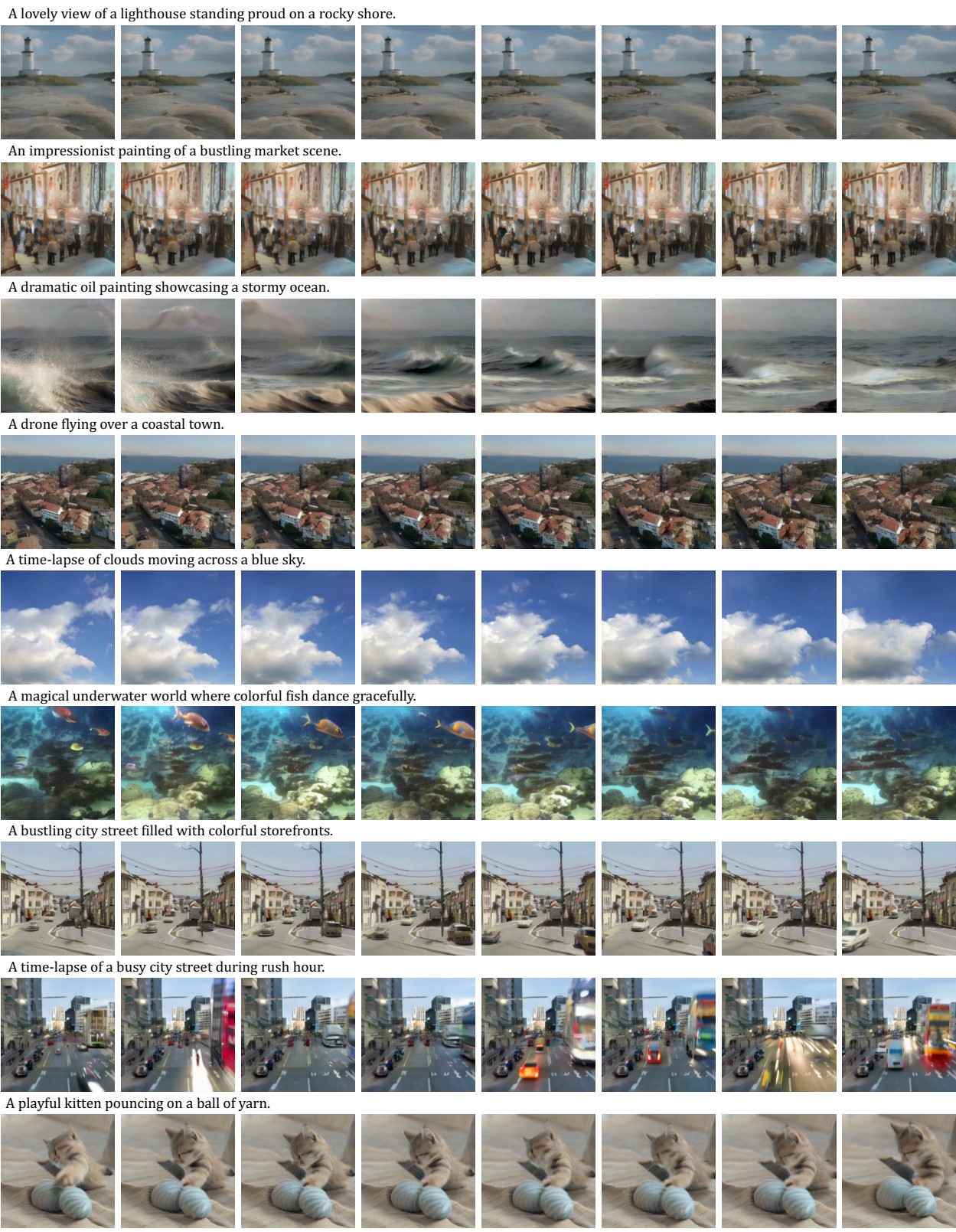

Figure 10: Additional video generation results.

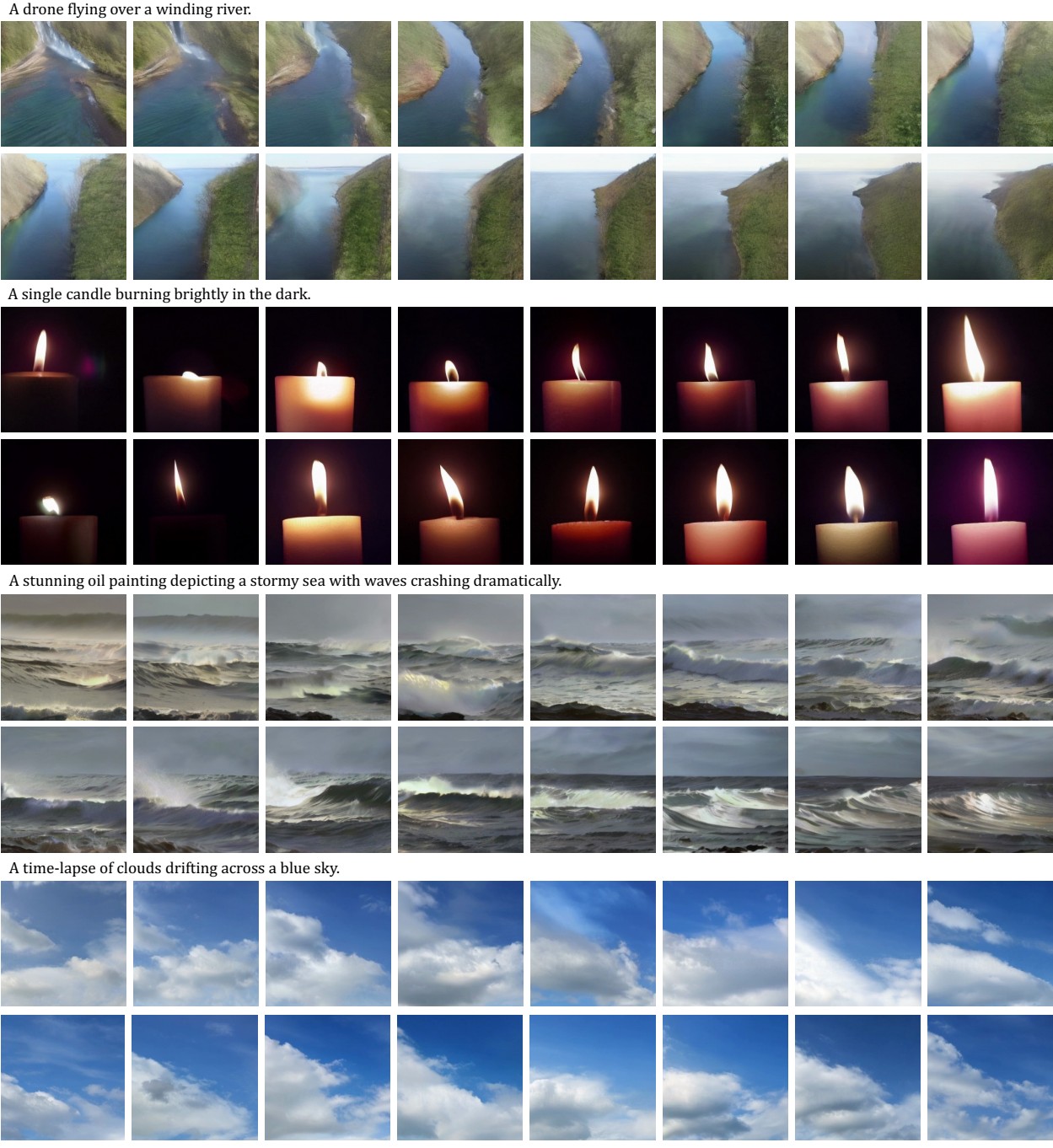

Figure 11: Additional long video generation results.

