# OpenReview forum: "DiCoDe: Diffusion-Compressed Deep Tokens for Autoregressive Video Generation with Language Models"
_TMLR — Under review for TMLR_

### Review · Reviewer_Tbgd · 2026-07-06

**Summary Of Contributions:**

1. This paper proposes DiCoDe, a two-stage framework that enables vanilla autoregressive (AR) large language models to generate long chronological videos by introducing diffusion-compressed continuous deep tokens.
2. Instead of low-resolution discrete VQ-VAE tokens (which produce massive token sequences and fail for long videos), the authors leverage a pre-trained video diffusion model as a powerful tokenizer. The model samples sparse endpoint frames along the video timeline, encodes each endpoint into only 16 continuous deep tokens via a CLIP encoder + query transformer. Adjacent video segments share endpoints, delivering an amortized ~1000× token reduction vs traditional VQ discrete tokens, eliminating the context window bottleneck for minute-length video generation.
3. The learned deep tokens are temporally causal (natively fit AR sequential modeling), extremely compact, and image-compatible—allowing the model to leverage abundant image-text datasets to mitigate scarce video-text training data. Short 2s video segments are fully reconstructed purely from the deep tokens of their start/end frames, exploiting temporal redundancy captured by diffusion priors.

**Audience:**

Yes

**Audience Explanation:**

TMLR’s readership spans theoretical machine learning, generative modeling, multimodal foundation models, autoregressive sequence modeling, and video synthesis researchers:
1. The core insight of compressing video into tiny continuous token sequences to run vanilla LLMs solves a critical limitation of text-only AR architectures for visual generation; the diffusion-as-tokenizer paradigm opens a new cross-modal integration direction distinct from VQ-based multimodal systems.
2. The ~1000× token compression addresses the dominant context window bottleneck plaguing minute-scale video autoregression, a heavily researched open problem in video generative models (Loong, VideoPoet, MAGI-1).
3. The GMM loss ablation and theoretical discussion of multi-modal uncertainty for continuous latent autoregression provides actionable design guidance for MAR/GIVT-style image/video AR models.
Diffusion model researchers: The work re-contextualizes pre-trained video diffusion models as high-level semantic compressors, a novel alternative to their standard use as pixel/latent generators; this dual-use perspective expands diffusion’s practical application scope.

**Broader Impact Concerns:**

1. DiCoDe enables lightweight, low-compute long video generation (up to 64s coherent clips) from arbitrary text prompts. Bad actors could generate realistic manipulated footage of events, people, or fictional news narratives without expensive GPU clusters, lowering the barrier for disinformation, political deepfakes, and harmful fabricated visual content.
2. The model learns visual semantics from millions of copyrighted photographs, artworks, and video clips without explicit creator consent. Generated outputs may inadvertently replicate copyrighted characters, artistic styles, or proprietary footage, creating legal and ethical IP risks for end users and model developers.

**Claims And Evidence:**

Yes

**Claims Explanation:**

1. Figure 1 provides direct numerical comparison between VQ-VAE low-level tokens and DiCoDe deep tokens, mathematically proving the ~1000× amortized compression rate. Tokenization reconstruction experiments (Figure 3) validate that 2s segments can be faithfully recovered from only two endpoint frames’ deep tokens across diverse motion types (object, camera, complex scenes). Zero-out ablation (Figure 4) demonstrates deep tokens encode high-level scene semantics rather than trivial pixel details, supporting the compression design rationale.
2. Table 4 ablation strictly isolates loss type as the only variable; FVD metrics clearly rank GMM (best) > single Gaussian > L2 (worst), with consistent quantitative evidence to justify multi-modal distribution modeling for continuous tokens.
3. Figure 8 plots training loss and MSR-VTT FVD across GPT-2 (117M–774M) and Llama 3.2 (1B–3B) model sizes. Larger models consistently yield lower loss and better video metrics, with clear separation between GPT and Llama families to confirm text knowledge transfers to visual sequence modeling.

**Requested Changes:**

1.Add formal analysis explaining why DiCoDe’s FVD/CLIPSIM lag fully optimized SOTA systems (VideoPoet, Show-1). Explicitly isolate confounding variables: training data scale (25M vs billions of image-text pairs), diffusion tokenizer lacking text conditioning, smaller AR backbone (1B Llama vs multi-billion proprietary models), limited training iterations. Add a controlled small-scale ablation against a simplified VideoPoet/Loong variant trained on identical 25M data to fairly disentangle compression tradeoffs from data/model scale effects.
2.Conduct pairwise human preference tests on short/long video clips across DiCoDe, Video-LaVIT, VILA-U baselines with standardized prompt set. Report aggregate human scores for metrics: motion dynamism, object consistency, prompt alignment, temporal coherence. Include results in a new table and supplement visual figures with statistical significance markers.
3.Repeat tokenizer training with a second open video diffusion model (e.g., VideoCrafter2) to verify the deep token compression pipeline is not overfit to DynamiCrafter priors. Report tokenization reconstruction quality and downstream zero-shot FVD to prove universal applicability of the diffusion-compression design.

---

### Review · Reviewer_Spqd · 2026-07-14

**Summary Of Contributions:**

## Summary

This paper proposes DiCoDe, a framework that treats videos as temporal sequences, analogous to language, and enables scalable video generation using an autoregressive language model.
The key idea is to use a video diffusion model as a high-compression tokenizer, representing an endpoint frame with 16 continuous deep tokens, and to train an AR language model to generate this sequence of deep tokens in temporal order. The authors use a GMM-based loss to model the conditional uncertainty of continuous tokens and claim that it outperforms an L2 loss. Through experiments on MSR-VTT zero-shot generation, tokenization reconstruction, short and long video generation, model scaling and ablations, the paper evaluates the compression efficiency and scalability of DiCoDe.

## Strengths
- The paper clearly motivates the research problem by emphasizing that videos are temporal sequences and that scalable video generation requires strong compression. It also points out the limitations of fixed-length diffusion clips and extremely long token contexts.
- The method is well structured: DiCoDe separates the diffusion tokenizer and the AR language model, guided by clear design requirements for deep tokens. This separation makes the overall pipeline easier to understand.
- The experiments show robust reconstruction quality in challenging cases and demonstrate applicability to both short and long video generation. The scaling and ablation studies further support the claims on scalability and efficiency.

## Weaknesses
- The paper is interesting as a system-level integration, but its conceptual novelty is limited. AR modeling of visual tokens, probabilistic prediction of continuous tokens, and diffusion-based video generation are all active existing directions. The main distinguishing element is the use of a video diffusion model as a high-compression tokenizer for endpoint deep tokens, but the experiments do not sufficiently isolate this design choice from plausible alternatives.
- The quantitative evaluation is limited. Although Tab. 2 reports CLIPSIM and FVD and shows competitive performance, the paper does not provide user studies, human evaluation, stress tests, or motion-specific metrics that would more directly assess the quality of temporal dynamics. Also, the paper lacks evidence of statistical reliability, such as variance across random seeds or confidence intervals.

**Audience:**

Yes

**Audience Explanation:**

The paper is likely to be of interest to readers working on video generation, long-video modeling, autoregressive visual generation, and compact visual representation learning.
Its main contribution is to connect video generation with aggressive token compression: instead of using a video diffusion model solely as a generator, DiCoDe uses it as a high-ratio tokenizer/compressor and delegates chronological modeling to an AR language model.
This perspective is relevant for researchers concerned with the context-length and computational bottlenecks of long-video generation.
That said, the paper's relevance would be clearer if it more clearly framed itself as a feasibility study of compressed AR video modeling and provided more controlled evidence for the quality-efficiency trade-off against alternative compression baselines.

**Broader Impact Concerns:**

The authors acknowledge the main broader impact concerns, including potential misuse for misleading content or deepfakes, copyright issues, and the propagation of dataset or model biases. The reviewer do not have additional broader impact concerns beyond those already discussed in the paper.

**Claims And Evidence:**

No

**Claims Explanation:**

The paper presents a technically plausible and reasonably convincing case that DiCoDe can generate videos under an aggressively compressed token representation.
The qualitative results on tokenization, short-video generation, and long-video generation support the claim that the method remains functional despite the high compression ratio.
However, the evidence is less conclusive for stronger claims about architectural novelty or superiority over alternative compact video representations.
The quantitative evaluation is relatively limited, mainly relying on CLIPSIM and FVD, and the baselines differ substantially in data scale, model scale, token budget, conditioning design, and evaluation protocol. Moreover, the paper lacks controlled comparisons against compression baselines under the same AR backbone, training data, and token budget.
Therefore, the main feasibility claim is supported, but stronger claims about novelty and quality-efficiency advantages should either be backed by additional controlled experiments or stated more conservatively.

**Requested Changes:**

- Report computational-efficiency metrics (Would strengthen the work)

Add a table reporting visual tokens per segment/second, AR context length, training GPU-days, inference latency, memory usage, sampling cost, and maximum supported video length across main baselines. This would strengthen the paper because DiCoDe's main claim is not only generation quality, but also efficient video modeling through aggressive token compression.

- Add token-budget-matched compression baselines (Critical for acceptance)

Compare DiCoDe against other compressed video representations under the same AR backbone, the same training data, and the same number of visual tokens. This is important to isolate the contribution of diffusion-compressed deep tokens and to distinguish it from the general benefit of using a compact.

- Provide more reproducibility details (Critical for acceptance)

Include detailed compute resources, GPU hours, training schedule, inference settings, token ordering, endpoint sharing procedure, or pseudo code for tokenizer training, AR training, and video generation. These details are needed to make the paper reproducible and to help readers to understand the interaction between the diffusion tokenizer and the AR model.

---

### Review · Reviewer_M9TY · 2026-07-16

**Summary Of Contributions:**

This paper proposes DiCoDe, a framework for autoregressive video generation using diffusion-compressed continuous deep tokens. The key idea is to use a video diffusion model as a high-ratio tokenizer/decoder, while an autoregressive language model predicts these tokens sequentially conditioned on a text prompt. The generated deep-token sequence is then decoded into video segments by the diffusion model. The paper further introduces probabilistic modeling of continuous deep tokens using Gaussian/GMM likelihoods, and reports experiments showing that the representation remains usable under aggressive compression and that larger AR models achieve better FVD and training loss.

**Strength:** The exploration of compact continuous tokens for AR video generation empirically shows that such tokens can support video synthesis under limited resources.

**Weakness:** The evidence mainly supports feasibility under aggressive compression, but is less conclusive about robust long-video generation and the semantics of the learned deep tokens.

**Audience:**

Yes

**Audience Explanation:**

The paper addresses an active and broadly relevant problem in generative modeling: how to make autoregressive video generation scalable despite the extremely long token sequences induced by visual data. Even if the current evidence is not sufficient to establish state-of-the-art visual quality, the findings are useful for understanding the compression-efficiency trade-off in AR video generation.

**Broader Impact Concerns:**

The paper has already included a Broader Impact statement. I do not see additional ethical issues.

**Claims And Evidence:**

Yes

**Claims Explanation:**

The main claim that DiCoDe can make highly compressed AR video modeling feasible is supported to a reasonable extent. The results show that the proposed representation does not collapse despite aggressive compression, and that larger AR backbones improve both loss and FVD.

However, some stronger implications are less convincingly supported:
1. The quantitative comparison in Table 2 should be interpreted mainly as a reference point rather than a controlled benchmark. The compared systems differ substantially. Thus, the table supports the usability of the compressed representation, but does not isolate the effect of the proposed method relative to prior work.
2. The method depends heavily on the diffusion tokenizer/decoder. This design is central to the paper and enables aggressive compression, but it also means that the learned token space and final video quality are constrained by the pretrained diffusion prior.
3. While the GMM likelihood is a reasonable modeling choice for continuous-token autoregression, its justification is mostly empirical. The paper shows that 16-GMM improves over L2 and a single Gaussian, but it does not systematically explore mixture sizes or alternative density models, nor does it establish that the learned deep-token distribution is well captured by a diagonal GMM.
4. The semantic nature of the deep tokens remains somewhat unclear. The zeroing-out experiment is suggestive, but it does not fully establish that individual tokens encode interpretable scene-level factors rather than simply providing generic conditioning information to the diffusion decoder.

**Requested Changes:**

**Critical changes:**

1. **Better evaluate robustness of the diffusion-compressed tokenizer.** Since the method relies on reconstructing short video segments from compact endpoint representations, the paper should test this assumption more systematically. The current qualitative reconstruction examples are useful but not sufficient to assess robustness.
2. **Ablate the temporal endpoint interval $f$.** The choice $f=16$, corresponding to a 17-frame 2-second segment at 8 fps, controls a central trade-off between compression ratio and reconstruction difficulty. The paper should evaluate sensitivity to this interval, or at least provide a stronger rationale for the default choice.

**Changes that would strengthen the work:**

1. **Provide a more detailed analysis of the learned deep-token space.** The zeroing-out experiment is interesting, but more systematic analyses would help substantiate the claim that deep tokens encode scene-level semantic factors.
2. **Strengthen the justification for GMM likelihood.** The paper should discuss more clearly why a 16-component diagonal GMM is chosen, whether the method is sensitive to the number of mixture components, and how it compares to alternative continuous density models. This is not necessarily required for the core contribution, but would make the probabilistic modeling component more convincing.
3. **Discuss computational cost more explicitly.** While the AR sequence is compact, generation still requires diffusion decoding for video segments. The paper would benefit from reporting inference time/cost and clarifying that the efficiency gain is primarily in AR temporal modeling rather than eliminating diffusion sampling cost.